# Pathological Interplay between Inflammation and Mitochondria Aggravates Glutamate Toxicity

**DOI:** 10.3390/ijms25042276

**Published:** 2024-02-14

**Authors:** Annette Vaglio-Garro, Andrey V. Kozlov, Yuliya D. Smirnova, Adelheid Weidinger

**Affiliations:** 1Ludwig Boltzmann Institute for Traumatology, The Research Center in Cooperation with AUVA, 1200 Vienna, Austria; annette.vaglio@trauma.lbg.ac.at (A.V.-G.); yuliya.dvoretskaya@trauma.lbg.ac.at (Y.D.S.); adelheid.weidinger@trauma.lbg.ac.at (A.W.); 2Austrian Cluster for Tissue Regeneration, 1200 Vienna, Austria; 3Laboratory of Metagenomics and Food Biotechnology, Voronezh State University of Engineering Technologies, 394036 Voronezh, Russia

**Keywords:** ferroptosis, glutamate, neuronal death, mitochondrial dysfunction, TCA cycle

## Abstract

Mitochondrial dysfunction and glutamate toxicity are associated with neural disorders, including brain trauma. A review of the literature suggests that toxic and transmission actions of neuronal glutamate are spatially and functionally separated. The transmission pathway utilizes synaptic GluN2A receptors, rapidly released pool of glutamate, evoked release of glutamate mediated by Synaptotagmin 1 and the amount of extracellular glutamate regulated by astrocytes. The toxic pathway utilizes extrasynaptic GluN2B receptors and a cytoplasmic pool of glutamate, which results from the spontaneous release of glutamate mediated by Synaptotagmin 7 and the neuronal 2-oxoglutarate dehydrogenase complex (OGDHC), a tricarboxylic acid (TCA) cycle enzyme. Additionally, the inhibition of OGDHC observed upon neuro-inflammation is due to an excessive release of reactive oxygen/nitrogen species by immune cells. The loss of OGDHC inhibits uptake of glutamate by mitochondria, thus facilitating its extracellular accumulation and stimulating toxic glutamate pathway without affecting transmission. High levels of extracellular glutamate lead to dysregulation of intracellular redox homeostasis and cause ferroptosis, excitotoxicity, and mitochondrial dysfunction. The latter affects the transmission pathway demanding high-energy supply and leading to cell death. Mitochondria aggravate glutamate toxicity due to impairments in the TCA cycle and become a victim of glutamate toxicity, which disrupts oxidative phosphorylation. Thus, therapies targeting the TCA cycle in neurological disorders may be more efficient than attempting to preserve mitochondrial oxidative phosphorylation.

## 1. Metabolism of Glutamate in Neuronal Tissue

Glutamate in the nervous system interfaces with many metabolic pathways, such as oxidation of energy, incorporation into proteins, and the formation of glutamine, gamma-aminobutyric acid (GABA), and glutathione. However, signal transmission via the glutamate-glutamine cycle is the major physiological function of glutamate in the nervous system [1] (Figure 1). Nevertheless, glutamate can also be toxic if occurring at high concentrations in extracellular fluid (Figure 1). Toxic extracellular concentrations of glutamate have been documented in a number of central nervous system (CNS) diseases [2] and brain injury [3]. There are two toxic pathways killing neurons upon elevated extracellular glutamate levels. The first one is excitotoxicity, which kills neurons via sustained activation of glutamate receptors and an excessive influx of Ca^2+^ into neurons [4]. The second, more recently described mechanism, ferroptosis, is a kind of programmed cell death mediated by iron [5]. A high concentration of glutamate in the extracellular fluid blocks glutamate/cysteine antiporters and reduces the intracellular levels of cysteine and, subsequently, the synthesis of glutathione. The latter inhibits glutathione peroxidase activity and induces neuronal death via iron-mediated oxidation of lipids [6].

In intact neurons, the major part of glutamate is packed into intracellular glutamate vesicles known to serve in signal transmission. Glutamate is released into the synaptic cleft upon activation of neurons in a process called evoked release. Images obtained by electron microscopy showed that astrocytes almost completely close to the synaptic cleft prevent the release of glutamate into the extracellular fluid [7], suggesting that only astrocytes located in close vicinity to synapses predominantly take up intrasynaptic glutamate. Astrocytes convert glutamate to glutamine, which is released into the extracellular fluid. Glutamine does not activate neuronal receptors, but neurons take up glutamine, convert it back to glutamate, and refill glutamate vesicles [8,9]. However, upon brain trauma and neurodegenerative diseases, the glutamate concentration in the extracellular fluid drastically increases due to the release of glutamate from damaged cells and from activated microglia and reactive astrocytes [10,11]. The distribution of glutamate in different compartments of the neuron is shown in Figure 2. The activation of glial cells always accompanies neuroinflammation, which occurs in most central nervous system (CNS) disorders [12]. In contrast to synapses, distant parts of the neuronal bodies have no tided contact with astrocytes. Subsequently, both neuronal and astrocytic bodies, which are not part of the synapsis structure, are exposed to extracellular glutamate [13,14,15]. Excessive activation of glutamate receptors in neurons causes their dysfunction and death [4]. There are several types of glutamate receptors described below but not all of them mediate neuronal death.

N-methyl-D-aspartate (NMDA) glutamate receptors can be formed by different types of isoforms: GluN2A, GluN2B, and GluN2C/2D. NMDA glutamate receptors with GluN2B and GluN2C/2D subunits are mainly located in the extrasynaptic site, whereas NMDA glutamate receptors rich in GluN2A are normally working in the signal transmission process located in the synaptic cleft. 

Under physiological conditions, glutamate will be packed in the so-called readily releasable pool. Under pathological conditions, for example after brain trauma, intra- and extracellular concentrations will change, leading to the diffusion of intracellular glutamate into the extracellular fluid and induction of excitotoxicity.

## 2. Heterogeneity of Glutamate Receptors

Several types of ionotropic glutamate receptors have been identified. They include ligand-gated ion channels called N-methyl-D-aspartate (NMDA) receptors, α-amino-3-hydroxy-5-methyl-4-isoxazolepropionic acid (AMPA) receptors, and Kainate receptors (KARs) [16,17,18]. All of them allow Na^+^ and K^+^ to flow through the plasma membrane of neurons upon activation [19]. Within these three types, only NMDA receptors, in addition to Na^+^ and K^+^, allow the entrance of Ca^2+^ into neurons. Inside neurons, Ca^2+^ activates numerous intracellular signaling cascades [20]; overwhelmed Ca^2+^ influx is the key factor causing excitotoxicity [21]. There are several types of NMDA receptors. All of them are tetramers with two glycine-binding subunits GluN1; two glutamate-binding subunits, such as GluN2A, GluN2B, or GluN2c/2D; as well as a combination of GluN2 and glycine-binding GluN3 subunits, such as GluN3A or GluN3B [22]. The major function of the NMDA receptors is to transmit the neuronal signal through the synaptic cleft. However, NMDA receptors are not only located at the synaptic cleft but also at the extrasynaptic site of neurons [23,24,25]. Synaptic and extrasynaptic NMDA receptors have different chemical structures. Synaptic receptors are enriched with the GluN2A subunit, while extrasynaptic receptors contain predominantly the GluN2B or GluN2C/2D subunits [25,26] (Figure 2). There is a body of evidence suggesting that extrasynaptic NMDA receptors containing GluN2B are primarily linked to excitotoxic effects of glutamate upon stroke and in a number of neurodegenerative diseases, as described in Table 1 [25,27,28], while activation of GluN2A subunits has a neuroprotective effect [29]. In general, the lowest agonist and co-agonist potency was found at GluN1/GluN2A NMDARs and the highest at GluN1/GluN2D NMDARs [30]. However, deactivation rates are fastest at GluN1/GluN2A NMDARs and slowest for GluN1/GluN2D NMDARs [30]. Similar deactivation rates were observed for NMDARs containing the GluN2B and GluN2C subunits [30]. These data suggest that transmission and the toxic effects of glutamate undergo different molecular pathways, resulting in the activation of either GluN2A or GluN2B, respectively.

Reasonably, in an attempt to prevent sustained activation of glutamate receptors, astrocytes and neurons should take up glutamate from the extracellular fluid. Glutamate transporter 1 (GLT-1) is the major transport system for glutamate in the CNS. Although GLT-1 was found predominantly in astrocytes, it is also expressed in neuronal bodies and axons [40]. The mechanisms mediating and regulating extracellular glutamate are not very clear. Astrocytes have a higher capacity to take up glutamate than neurons, but since only local glutamate levels close to neurons induce excitotoxicity, the uptake of glutamate by neurons may be critical for glutamate toxicity. The further fate of glutamate in neurons and astrocytes is also different.

## 3. Intracellular Metabolism of Glutamate

In the majority of cells, including neurons, intracellular glutamate is predominantly used for ATP synthesis and converted into other organic molecules. One specific function of glutamate in neurons is the activation of postsynaptic receptors, playing a central role in neuronal signal transduction. In contrast, the major fate of glutamate in astrocytes is its conversion into glutamine [41,42,43,44,45] (Figure 2). It has been unequivocally accepted that astrocytes do not require glutamate and aerobic metabolism through mitochondria to meet their bioenergetics demands [46]. Glutamate is converted to glutamine (GLN) by glutamine synthetase (GS). GS is expressed exclusively in astrocytes, which is an important segment of the glutamate-glutamine cycle [47]. Thus, the glutamate consumed by astrocytes will not be extracted from the total glutamate pool, and it will, again and again, reappear in neurons as glutamine, which will be converted back to glutamate [1]; the latter can theoretically contribute to both signal transduction and toxicity. Being a part of blood–brain barrier, astrocytes take up glucose from the blood, convert it to pyruvate and then to lactate by glycolysis. Lactate will be transported to neurons (lactate shuttle) and transformed in neurons back to pyruvate, which will be used as a mitochondrial substrate to provide ATP [48].

In contrast to astrocytes, neurons strictlyy require mitochondria with active oxidative phosphorylation (OxPhos) and the tricarboxylic acid cycle (TCA-cycle). They have higher energetic demands to support extensive ion flows accompanying the activation of neurons and signal transmission and, consequently, mitochondria play a key role in the energy metabolism in neurons [49,50]. This suggests that neurons, rather than astrocytes, can burn glutamate in mitochondria, subsequently extracting it from the glutamate-glutamine pool [51,52], thus preventing glutamate toxicity. Divakaruni et al. reported the first strong experimental evidence supporting this assumption. The authors inhibited the pyruvate transporter in neurons and reported three important observations: the inhibition of the mitochondrial pyruvate transporter in primary cortical cultures (1) shifted neuronal metabolism in favor of glutamate consumption, (2) increased the consumption of glutamate in neurons, which caused a drop in the extracellular glutamate levels, and (3) reduced cell death rate [53]. This study shed new light on how neurons can cope with glutamate excitotoxicity, namely increasing the rate of glutamate oxidation in mitochondria. Logically, one can therefore expect that the inhibition of glutamate uptake by mitochondria causes the opposite effect, i.e., an increase in extracellular glutamate levels and aggravation of glutamate toxicity. 

Indeed, a recent publication by Weidinger et al. [54] investigated the segment of the TCA cycle responsible for glutamate uptake in mitochondria and showed that inhibition of this TCA cycle segment in neurons resulted in an increase of extracellular glutamate levels and elevated neuron death rate, which is in line with Divakaruni et al. [53]. In fact, indirect evidence of the involvement of the glutamate segment of the TCA cycle in glutamate toxicity was already previously reported under different pathological situations. A remarkable inhibition of mitochondrial respiration by glutamate was observed in the cortex of rats subjected to traumatic brain injury (TBI), while pyruvate respiration was affected much less [55], suggesting a defect at either the glutamate segment of the TCA cycle or complex I of the respiratory chain of mitochondria. A follow-up study showed that the mechanical injury of the cortex, occurring upon TBI, resulted in the inhibition of the oxoglutarate dehydrogenase complex enzyme (OGDHC), one out of two enzymes responsible for the uptake of glutamate in mitochondria [56]. The oxidation of glutamate in the mitochondria requires glutamate dehydrogenase (GDH) and OGDHC. However, in contrast to OGDHC, no changes in GDH activity were found upon TBI, suggesting that OGDHC is more susceptible to inhibition upon brain injury [54], although a decrease in GDH activity was reported in a penetrating ballistic-like brain injury model [57]. Other data shed light on the possible mechanism underlying the inhibition of OGDHC upon brain injury. It has been shown that the inhibition of glutamate uptake by mitochondria coincides with increased cortical nitric oxide (NO) levels [55]. A similar relationship between brain NO and glutamate levels was observed in patients with subarachnoid hemorrhages (SAHs) [58]. The clinical status of these patients was monitored by examination of microdialysate obtained from brain tissue close to the injured area. It was shown that, in the acute phase of SAH, these patients manifested elevated levels of glutamate, which were accompanied by a substantial increase in the levels of NO in microdialysate [58]. The assumption that NO has a critical role is in line with the fact that there are many sources of NO in the brain, which can be activated upon CNS disorders, such as brain injury [59], Parkinson’s disease, and Alzheimer’s disease [60]. A body of literature suggests that NO plays important physiologic [61,62] as well as pathologic [63] roles in CNS disorders. It has been assumed that the pathologic role of NO is due to S-nitrosylation of proteins, which lose their structure and function, contributing to the development of neurodegenerative diseases [64,65,66]. The majority of the existing reports on the pathologic impact of NO in CNS disorders consider reversible and/or irreversible inhibition of the mitochondrial respiratory chain by NO and its derivatives, impairing the bioenergetics status of neurons [67,68]. However, recently, Weidinger et al., have shown that the most sensitive target of NO in mitochondria is not the electron transport chain (ETC), but OGDHC [54]. Weidinger et al. showed that NO, at concentrations previously determined in vivo [55], inhibits OGDHC rather than ETC. Of note, the inhibition of OGDHC does not necessarily cause inhibition of the ETC because mitochondrial respiration and ATP synthesis can be recovered by switching to other substrates or bridging the electron transport via extra-mitochondrial pathways [69,70,71]. In contrast, glutamate uptake cannot be bridged by other pathways and fully depends on the activity of OGDHC. This leads us to the fundamental assumption that mitochondria play a boomerang-like role in the CNS. They induce glutamate toxicity, slowing down the oxidation of glutamate, but later they suffer from glutamate toxicity, reducing respiratory activity and ATP synthesis. The fact that the drop in bioenergetics capacity of mitochondria is secondary to glutamate toxicity has already been supported by a body of literature [72,73,74]. The fact that the primary mitochondrial defect is inhibition of mitochondrial OGDHC, but not a defect at the ETC level, substantially changes the view on therapeutic strategies.

In contrast to the defects occurring in the ETC, which can be hardly recovered (perhaps only genetically), the activity of OGDHC is easier to compensate by using thiamine (vitamin B1), a cofactor of this enzyme. Indeed, it has been shown that treatment with thiamine restored OGDHC activity and mitochondrial respiration in rats impaired by TBI [56]. A similar relationship between NO levels, OGDHC activity, and glutamate toxicity was already observed in patients with Wernicke–Korsakow syndrome. This CNS disorder is characterized by thiamine deficiency and can be efficiently reversed by thiamine substitution [75,76]. This suggests thiamine is a valid therapeutic option in humans. Of note, the activities of pyruvate dehydrogenase (PDH), another TCA cycle enzyme, which has thiamine as a cofactor, and GDH (the second enzyme contributing to glutamate uptake) did not change in the experimental models described above [54,56], suggesting the high vulnerability of OGDHC. These data suggest that the OGDHC of neurons attenuates extracellular glutamate levels and glutamate toxicity. Below, we consider how changes in OGDHC activity can regulate the levels of extracellular glutamate.

In humans, glutamate toxicity can be addressed by the inhibition of ionotropic glutamate receptors or by the decrease of extracellular glutamate. Magnesium is one of the most common inhibitors of Ca^2+^ flux through NMDA receptors, meaning that in its presence, glutamate can only act on the AMPA receptor [77,78]. Some other blockers of glutamate receptors are used in preclinical models [38,79]. The data analyzed in this review address the reduction of extracellular glutamate by re-activation of OGDHC by its cofactor, thiamine, which is used in clinical practice for the treatment of Wernicke–Korsakov syndrome. Our data suggest that thiamine is a valid therapeutic option for all neurodegenerative diseases and brain injury in humans [54,56].

## 4. Mechanisms Regulating Glutamate Absorption in Neurons

GLT-1 is the major glutamate transport system, which accounts for approx. 95% of the total glutamate uptake activity in the brain tissue [80,81,82]. GLT-1 co-transports one glutamate with 3 Na^+^ and 1 H^+^ and with a counter transport of 1 K^+^ into the extracellular fluid per cycle (Figure 3). GLT-1 is highly expressed in astrocytes but also in neurons. However, the function of neuronal GLT-1 is relatively unexplored [40,83]. There is evidence suggesting that, in neurons, GLT-1-mediated glutamate uptake provides glutamate predominantly for energy metabolism [83] (in contrast to the glutamine-glutamate cycle operating in close vicinity to synapses). Perhaps, extracellular glutamate can be located on the body of neurons far from synapses and be absorbed by GLT-1, which facilitates its uptake by mitochondria before it diffuses to synapses through the axon. GLT-1 transport can be regulated, but the regulation mechanisms have been investigated predominantly in astrocytes because they carry most of this transporter in the body; similar mechanisms can also operate in neurons [40,84]. It has been shown that the rate of glutamate transport in astrocytes can be regulated by GLT-1 expression/activity and by their lateral diffusion to specific locations with high glutamate levels [85]. The change in the rate of glutamate absorption by GLT-1 can regulate not only its toxic effects but also the magnitude and duration of the activation of postsynaptic neurons [85,86]. Also, approx. 5% of non-GLT-1 transporters regulate extracellular glutamate persistence and the duration of NMDA receptor activation [87]. The rate of glutamate uptake in astrocytes is substantially higher than in neurons (approx. 4 nmol Glu/min/mg protein) [88] vs. approx. 0.2 nmol Glu/min/mg protein as recalculated based on data in Rink et al. [89]. However, since neuron glutamate receptors are responsible for the induction of toxic effects, the regulation of glutamate concentration close to the neuronal plasma membrane by neuronal transporters can be important. 

Theoretically, since an increased concentration of extracellular glutamate is often due to tissue damage accompanied by the release of K^+^, high levels of extracellular K^+^, and high concentration of intracellular glutamate, if this compound is not taken up by mitochondria and vesicles, this process can slow down the absorption of glutamate and increase glutamate toxicity. This, however, can be compensated by an increased number of GTL-1 by moving them to the spot with high glutamate concentration [86]. However, there is no clear data on how the increase in intracellular glutamate levels can influence the flow of glutamate through GLT-1. This mechanism can be linked to the intracellular distribution of glutamate between different intracellular pools. Rimmele et al. reported an interesting observation. They showed that the conditional knockout (KO) of GTL-1 in neuronal axons (predominantly located in presynaptic terminals) induces excitotoxic death of neurons, while it does not affect the stimulus of evoked glutamate release [40]. This suggests that two different pools of glutamate are responsible for the transmission and toxic effects of glutamate and that neurons contribute to the clearance of the toxic glutamate pool. Moreover, electron microscopy combined with immunolabeling showed that GTL-1 extends beyond the terminal portion of axons [40]. This suggests that neurons, via their GTL-1 transporters, are able to clean up the extrasynaptic glutamate pool responsible for toxic effects. This, however, still does not bring a link involving OGDHC in the regulation of glutamate toxicity. The key to understanding this link comes by considering the intracellular destinations of glutamate.

Glutamate has several different purposes in the body: (1) In mitochondria, it can be used as a source of energy when it is fed into the tricarboxylic acid cycle. (2) In the cytoplasm, it can be packed in vesicles (resting pool) and reach different fates. For example, in the synaptic cleft, the membrane of the vesicles of the readily releasable pool fuses with the cell membrane by interacting with Synaptotagmin 1 (Syt1) to form the synapsis structure and release the glutamate (2a). Additionally, at the extrasynaptic site, the membrane of the vesicles of the resting pool interacts with Synaptotagmin 7 (Syt7) of the cell membrane to release glutamate in the extracellular fluid outside of the synaptic cleft (2b). The latter can interact with some glutamate transporters, like GLUT1, as a returning pathway into the intracellular fluid or activate extrasynaptic NMDA glutamate receptors. Here, Km stands for the half-saturation transport constant, which is also known as the apparent affinity constant (Ka).

## 5. Mechanisms Regulating the Balance between Intracellular and Extracellular Glutamate Pools

After entering the neuron, glutamate can undergo two major pathways, i.e., energy conservation in mitochondria and signal transmission via synaptic vesicles (Figure 3). The distribution of glutamate between these two pathways is regulated based on their relative affinities to the mitochondrial and the vesicular glutamate transporters (VGLUTs). The mitochondrial transporter has a much higher affinity to glutamate [*K*_m_ = 0.2 mM [90]] than VGLUT [*K*_m_ = 2 mM [91]]. Following this fact, glutamate entering neurons through GTL-1 will be predominantly taken up by mitochondria and used for ATP synthesis. However, if mitochondria do not efficiently consume glutamate, glutamate will enter glutamate vesicles. However, the vesicle pool in neurons is not homogeneous and the fate of glutamate depends on the vesicle pool it enters. There are three pools of glutamate vesicles in neurons: (1) the readily releasable pool (RRP), (2) the recycling pool, and (3) the reserve pool [92]. There are two types of glutamate release from vesicles, evoked (stimulated, including stimulation with sucrose) [93] and spontaneous [94,95,96]. The evoked release is well known and serves for neuronal signal transmission, while spontaneous release is less explored. It is known that the spontaneous release plays a physiological role in supporting synaptic maturation and homeostasis (for review, see [97]) but it is also associated with glutamate excitotoxicity [98]. It has been shown that evoked glutamate release originates from RRP, while spontaneous release originates from the resting pool [97,99]. In contrast to evoked release in the synapse, the spontaneous glutamate release occurs in the axon membrane located out of the active zone of synapses [99]. The spontaneous release of glutamate, which is mediated by extrasynaptic glutamate receptors, is linked to the toxic effects of glutamate. For example, the elevated spontaneous release is associated with upregulation of VGLUT, thus causing an increase in intravesicular glutamate levels and suggesting that neurons attempt to release an excess of glutamate in extracellular fluid, which may multiply the toxic effects of glutamate. Genetic upregulation of VGLUT facilitates glutamate transport into vesicles but simultaneously increases spontaneous release of glutamate from these vesicles before they enter RRP and induces excitotoxic effects on neurons [98]. Evoked and spontaneous glutamate release are regulated by different mechanisms (Figure 3). The release of glutamate from vesicles is regulated by synaptotagmin, a family of Ca^2+^ sensor proteins regulating the release of glutamate from the vesicles and hormone secretion. It has been shown that the evoked release of glutamate is regulated by Syt1, which simultaneously inhibits the spontaneous glutamate release [100], while Syt7 triggers spontaneous glutamate release and efficiently activates GluN2B glutamate receptors [101] (Figure 3). Action potential-evoked neurotransmitter release is associated with the transduction of neuronal signals, while spontaneous release has been associated with neurological disorders. Also, under certain circumstances, glutamate that is released in the synapses can diffuse out of the synaptic cleft and activate nearby extrasynaptic glutamate receptors [102]. Collectively, these results support the premise that spontaneous and evoked neurotransmission activate distinct sets of NMDA receptors, which are responsible for neurotransmission and toxic effects, respectively [103]. Below, we show the list of evidence suggesting that transmission and toxic effects of glutamate undergo different pathways, i.e., intrasynaptic and extrasynaptic, respectively (Table 2). We assume that both pathways either do not communicate or have only limited communication between each other. There is a logical biological explanation for this observation. By separating these two pathways, a neuron preserves its transmission function; when the extracellular concentration of glutamate overpasses the threshold and the neuron cannot offset anymore the process, the glutamate_IN_-glutamate_OUT_ cycle will be activated to induce neuronal death. In this way, single neurons will die before the transmission function is seriously impaired. OGDHC plays a critical role in turning on the vicious glutamate_IN_-glutamate_OUT_ cycle, reducing the consumption of extracellular glutamate and re-directing an excess of intracellular glutamate back to the extracellular pool via spontaneous release (Figure 3).

## 6. Glutamate-Mediated Death of Neurons

Glutamate can induce different death pathways. Primary pathways include excitotoxicity [18] and ferroptosis [109,110]. The excitotoxic pathway is characterized by excessive excitation of neurons and influx of Ca^2+^ into the cell. Ca^2+^ activates different enzymatic systems, such as proteases, phospholipases, NOS, and other Ca^2+^-dependent enzymes. An excess of Ca^2+^ also stimulates the formation of mitochondrial permeability transition pores (mPTPs) and subsequent mitochondrial dysfunction accompanied by reduced ATP levels and damage to the mitochondrial membrane [74] (Figure 4). The damage of mitochondrial membranes facilitates the release of cytochrome c and the induction of apoptosis (Figure 4). More recently, it has also been shown that ferroptotic cell death can be activated. High levels of extracellular glutamate block the glutamate cysteine antiporter, causing the depletion of intracellular cysteine and then glutathione. The latter inhibits glutathione peroxidases and induces oxidation of lipids and ferroptosis [110], resulting in iron-dependent non-apoptotic cell death [110]. The Xc-antiporter system consists of a light chain subunit SLC7A11 and heavy chain subunits SLC3A2. Typically, stress-induced SLC7A11 expression is mediated by two major transcription factors: activating transcription factor 4 (ATF4) and/or nuclear factor erythroid 2-related factor 2 (NRF2). However, recently, a study by Zhang et al. showed that erastin (a ferroptosis-inducing agent) induces SLC7A11 expression through an NRF2- or ATF4-independent mechanism [111]. It has also been shown that p53 is able to inhibit cystine uptake by repressing SLC7A11 transcription, which limits the production of intracellular glutathione and increases the sensitivity of cells to ferroptosis [112]. All this indicates a critical role for SLC7A11-mediated cystine uptake in the suppression of ferroptosis, including in neuronal cells.

Fe^2+^ concentrations and expression of genes associated with ferroptosis are altered during intracerebral hemorrhages. The potential antioxidant agent crocin has been shown to inhibit neuronal cell ferroptosis through the expression of Nrf2 and SLC7A11, thereby alleviating intracerebral hemorrhages [113]. In an oligodendrocyte model of RSL3-induced ferroptosis, it was shown that RSL3 inhibits the main anti-ferroptosis pathway, i.e., the SLC7A11/glutathione/glutathione peroxidase 4 pathway [114]. A study by Zhao et al. on mouse hippocampal HT-22 cells treated with ferric ammonium citrate showed that Isorhynchophylline (alkaloid isolated from *Uncaria rhynchophylla*) is able to protect neurocytes from ferroptosis through the miR-122-5p/TP53/SLC7A11 pathway, which could become a potential therapeutic target for intracerebral hemorrhages [115]. Another study showed that oxygen-glucose deprivation/reperfusion (OGD/R)-induced neuronal injury induces ferroptosis by reducing the levels of SLC7A11 and glutathione peroxidase 4 (GPX4) and also reduces the levels of endogenous antioxidants, including NADPH, glutathione, and superoxide dismutase in neurons [116].

All these data point to the involvement of SLC7A11 and ferroptosis in the pathogenesis of many neurological diseases and suggest ferroptosis as an attractive therapeutic target.

## 7. Interplay between Glutamate Mediated Excitation and Intracellular Glutamate Signaling

Above, we discussed the toxic action of glutamate mediated by glutamate receptors and transporters. However, additionally, glutamate is also involved in a number of intracellular signaling cascades, which can interfere with the action of glutamate described above. Previously mentioned mechanisms include the action of ionotropic glutamate receptors, which are ligand-gated transmembrane channels that selectively transport cations. This not only induces action potential but also impacts several intracellular signaling cascades. In neurons, ionotropic glutamate receptors-mediated calcium influx activates Src family kinases, PI3K and ERK [117,118]. In addition to ionotropic glutamate receptors, neurons and a number of glial cells express three groups of metabotropic glutamate receptors (mGluRs). mGluRs are members of the G-protein-coupled receptor (GPCR) superfamily and they are coupled to inositol trisphosphate (group I) IP3 or to cAMP (group II and III), mediating intracellular signaling pathways. Group I mGluRs activate a range of molecular pathways, including phospholipase D and several protein kinases. Group II and III inhibit adenylyl cyclase and permeability of Ca^2+^ channels. Additionally, group III stimulates cGMP-dependent pathways [119]. Group I mGluRs are often localized postsynaptically and, in general, increase neuronal excitability, while group II and group III are localized on presynaptic axons and inhibit neurotransmitter release and excitation. In addition, the neurotransmitter release causes a relatively short but strong acidification of the synaptic cleft [120,121]. In addition, glutamine, which is synthesized during the glutamate–glutamine cycle, is a very important amino acid for the regulation of the acid–base cellular equilibrium [122,123]. These multiple intracellular pathways of glutamate can modulate the excitation, neurosignal transmission, as well as death cell pathways operating in neurons upon high levels of extracellular glutamate. We expect that the interaction between these three major functions of glutamate, neuronal signal transmission, toxicity, and intracellular signaling pathways will be an important area of research in the following years.

## 8. Conclusions

Glutamate toxicity plays a pivotal role in brain injury and neurodegenerative diseases. An excess of glutamate kills neurons via excitotoxic and ferroptotic pathways. The recent literature shed light on the mechanisms controlling glutamate-mediated neuronal death. The toxic effects of glutamate, at least partially, are controlled by OGDHC, a TCA cycle enzyme in neurons. Interestingly, insufficiency of this enzyme predominantly influences the turnover of extracellular glutamate but not OxPhos. Impaired OxPhos, a common feature of neuronal diseases, appears to be secondary to glutamate toxicity. This toxic effect is spatially and functionally separated from the transmission function of glutamate. This suggests that glutamate toxicity can kill neurons without affecting the transmission function. In contrast to the respiratory function of mitochondria, which cannot be easily improved, the activity of OGDHC can be recovered by treatment with its cofactor thiamine. We assume that there is only a very weak communication between the transmission and the toxic pathways of glutamate, aiming to preserve transmission function in toxic environments. 

## Figures and Tables

**Figure 1 ijms-25-02276-f001:**
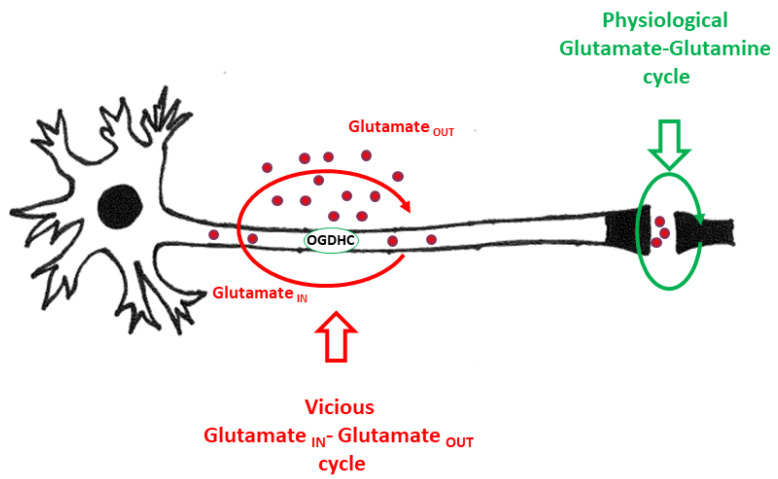
Graphical abstract. The glutamateglutamate vicious toxic cycle (shown in red) occurs at the extrasynaptic site. Physiological glutamate-glutamine cycle (shown in green) is located in the synaptic cleft.

**Figure 2 ijms-25-02276-f002:**
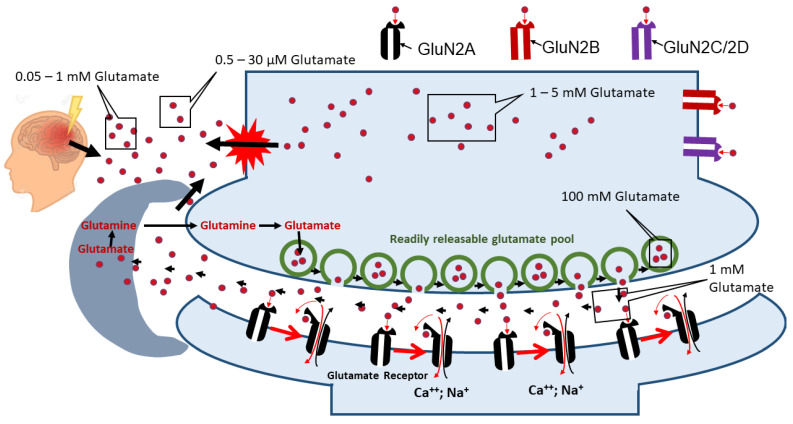
Distribution of glutamate in the nervous tissue, and the glutamate-glutamine cycle and its changes upon brain injury.

**Figure 3 ijms-25-02276-f003:**
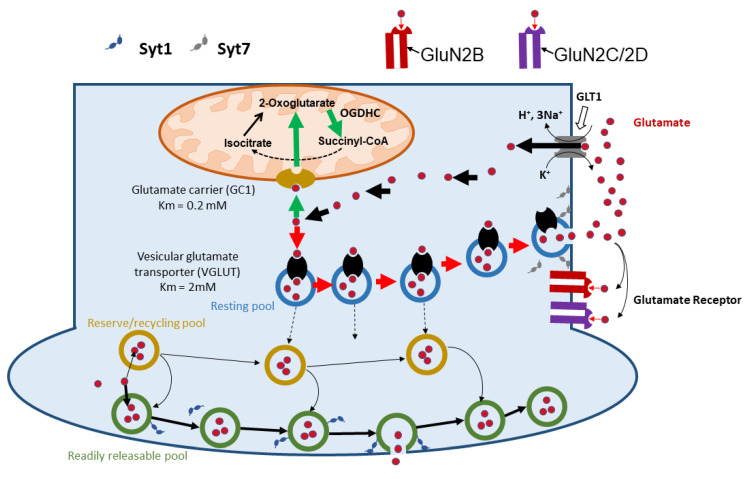
Evoked and spontaneous glutamate release, resulting in a balance between extracellular and intracellular glutamate pools. Spontaneous release and uptake of glutamate by GLT1 create a vicious Glutamate_in_-Glutamate_out_ cycle.

**Figure 4 ijms-25-02276-f004:**
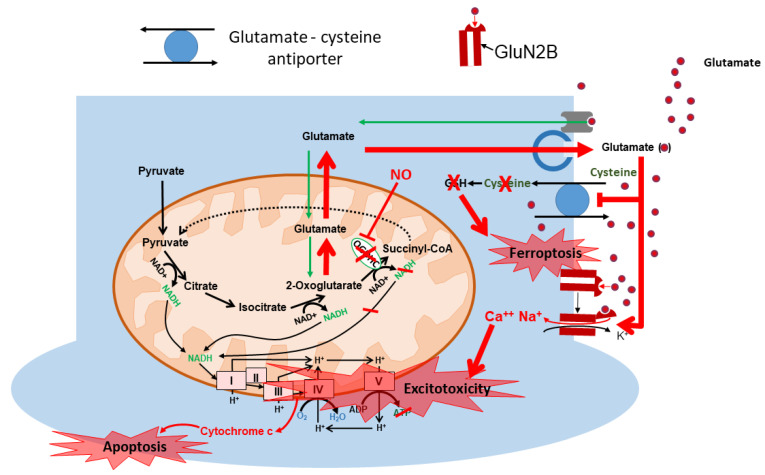
**The impact of mitochondria in mechanisms of glutamate-mediated neuronal death.** When the oxoglutarate dehydrogenase enzyme is blocked in mitochondria, the glutamate accumulates. Inside the mitochondria, it can disrupt the normal function of the electron transport chain, where the remaining glutamate will leak into the cytoplasm and secreted into the extracellular fluid. The high concentration of glutamate in the extracellular fluid will affect other transporters like (1) the glutamate-cysteine antiporter, leading to the reduction of intracellular cysteine, and thus reducing the activity of enzymes of the family of the glutathione peroxidase, causing the accumulation of lipid peroxides that, together with iron, can cause ferroptosis. Additionally, (2) glutamate can cause excitotoxicity via activated extrasynaptically NMDA receptors.

**Table 1 ijms-25-02276-t001:** Examples of neurological disorders related to glutamate toxicity.

Pathology	Mechanism	Reference
Amyotrophic lateral sclerosis (ALS)	Increased glutamatergic neurotransmission is present in ALS and might contribute to neurodegeneration. Downregulation of EAAT2 and upregulation of system Xc cause increased activation of glutamate receptors.	[31,32,33]
2. Alzheimer’s disease (AD)	Increased glutamate concentrations, changes in calcium homeostasis, and increased sensitivity of NMDA receptors make neurons more sensitive to excitotoxicity in AD.	[34,35,36]
3. Parkinson’s disease	Impaired mitochondrial function can make the substantia nigra sensitive to glutamate.	[37]
4. Huntington’s disease	In HD, there is a redistribution of NMDA receptors to the extrasynaptic compartment, which might activate signaling pathways that foster excitotoxicity and neurodegeneration.	[38]
5. Multiple sclerosis	Glutamate excitotoxicity mediated by the AMPA/Kainate type of glutamate receptors is an important mechanism in autoimmune demyelination.	[39]

**Table 2 ijms-25-02276-t002:** Evidence suggesting that transmission and toxic effects of glutamate undergo spatially and functionally separated pathways.

Evidence	Reference
Transmission and toxic pathways engage NMDA receptors characterized by two different chemical structures, GluN2A and GluN2B, respectively.	[22]
2. Transmission and toxic pathways engage NMDA receptors characterized by two different locations, synaptic and extra-synaptic, respectively.	[23,24,25]
3. Glutamate absorbed by neurons enters the cytoplasmic pool and is used by mitochondria rather than packed in RRP vesicles.	[97,99]
4. Transmission and toxic pathways engage two different pools of vesicles, RRP and reserve pool, respectively.	[40]
5. Transmission and toxic pathways are mediated by two different pathways of glutamate release evoked and spontaneous, respectively.	[93,94,95,96]
6. Evoked and spontaneous glutamate release are regulated by two different mechanisms, mediated by Syt1 and Syt7, respectively.	[100,101]
7. Evoked and spontaneous glutamate release activate two different pools of glutamate receptors, intra- and extra-synaptic NMDA receptors, respectively.	[98,99,103]
8. The activity of OGDHC balances intra- and extra-cellular glutamate levels and glutamate toxicity.	[54,104]
9. Low OGDHC activity causes an increase in extracellular glutamate and cell death, while high activity reduces the levels of extracellular glutamate and cell death rate.	[105,106,107,108]

## Data Availability

No new data were created or analyzed in this study. Data sharing is not applicable to this article.

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
