# Peer review of "Pathological Interplay between Inflammation and Mitochondria Aggravates Glutamate Toxicity"

_ijms, 2024, doi:10.3390/ijms25042276_

Round 1
Reviewer 1 Report
Comments and Suggestions for Authors
All the figs should be high resolution figs.
Some of the references are missing in the following paragraph.
In contrast to astrocytes, neurons obligatory require mitochondria with active oxida tive phosphorylation (OxPhos) and tricarboxylic acid cycle (TCA-cycle). They have higher energetic demands to support extensive ion flows accompanying activation of neurons and signal transmission, and consequently mitochondria play a key role in the energy 136 metabolism in neurons. This suggests that neurons, rather than astrocytes, can burn glutamate in mitochondria, subsequently extracting it from the glutamate-glutamine pool, 138 and prevent glutamate toxicity.
Provide a table where glutamate toxicity leads to diseases.
Please write about how could glutamate toxicity be addressed in humans?
Reviewer 2 Report
Comments and Suggestions for Authors
There are two main issues:
1) What is the role of glutamate cell signaling? The authors have to show that in the text under a separate title. Also, the authors have to show the importance of glutamate in the acid-base balance.
2) What is the Km value in Figure 2? Km is an important enzyme constant, and it is suggested to use another expression other than Km and to express it as Ka, which is the abbreviation of affinity constant.
3) Figure quality should be improved. Are the figures designed by the authors without the involvement of a third party?
Other issues:
Is the first figure a graphical abstract? It is not numbered, and it is better to denote that it is a graphical abstract.
Comments on the Quality of English Language
Minor revision of English is required.
Round 2
Reviewer 2 Report
Comments and Suggestions for Authors
The authors showed a good response.
Comments on the Quality of English LanguageMinor editing is required.